# The Legacy of Hg Contamination in a Past Mining Area (Tuscany, Italy): Hg Speciation and Health Risk Assessment

**DOI:** 10.3390/toxics12060436

**Published:** 2024-06-16

**Authors:** Simone Arrighi, Fabrizio Franceschini, Riccardo Petrini, Silvia Fornasaro, Lisa Ghezzi

**Affiliations:** 1Department of Earth Science, University of Pisa, Via S. Maria 53, 56126 Pisa, Italy; simone.arrighi@unipi.it (S.A.); riccardo.petrini@unipi.it (R.P.); lisa.ghezzi@unipi.it (L.G.); 2Environmental Protection Agency of Tuscany (ARPAT), Via Vittorio Veneto, 56127 Pisa, Italy; f.franceschini@arpat.toscana.it

**Keywords:** chlor-alkali plant, mercury legacy, Tuscany, risk assessment, Hg speciation

## Abstract

The mercury cell manufacturing process, which has been extensively used in chlor-alkali plants to produce chlorine and caustic soda by electrolysis, represents a major source of Hg environmental pollution. At Saline di Volterra (Tuscany, Italy), solution mining by pumping water into halite deposits was applied to produce brines for a mercury-cell chlor-alkali plant. The Hg-contaminated, exhausted brines were pumped back at depth into the rock salt field in order to renew the available resources. Activities ceased in 1994, following the leakage dispersion of highly contaminated Hg(0)-bearing brines into the environment. The mercury content in the soil, measured during a survey conducted in 2000, reached 334 mg/kg, highlighting diffuse contamination in the floodplain. By 2009, the Hg concentration had generally decreased and was mostly confined to the topsoil layer. In order to evaluate the present Hg soil pollution, a geochemical survey was carried out in 2023, almost thirty years after the contamination event. The obtained data indicated the occurrence of legacy Hg, which reached 25.5 mg/kg in some soil samples. Speciation analysis for the most contaminated soil revealed that Hg(0) represented about 17.3% of the total Hg and that water-soluble and organic Hg fractions were negligible. These results suggest that the originally released metallic mercury has volatilized and likely oxidized, becoming practically immobile in the soil. A risk assessment, performed by applying Hg speciation analysis, indicated that the mercury in the soil does not carry a risk of non-cancerous effects for different exposure routes in case of subsequent use of the site and that the formerly contaminated area can now be converted into a leisure area.

## 1. Introduction

Mercury is a highly hazardous heavy metal for humans and ecosystems [1,2,3]. It is included among the chemicals of major public health concern [4,5], posing a risk to public health at a global level, particularly when it accumulates in the soil [6]. Mercury may exist in ionic, organic, and elemental forms. Humans experience most of their Hg exposure when eating fishery products contaminated by methylmercury (MeHg), a potent neurotoxin [7]. However, exposure may also occur through contaminated soil via ingestion or dermal contact, as well as by inhalation of Hg-bearing particulate matter and Hg vapor [8]. Mercury is released into the environment from both geogenic sources and anthropogenic industrial processes, with the latter including gold mining [9], stationary fuel combustion [10], oil refinement [11], cement production, non-ferrous metal manufacturing, and pig iron and steel production [12,13], as well as waste incineration and chlor-alkali plants (CAP) [14,15]. Mercury chloride has also been used as a biocide in conservation treatments until its use was highly restricted or forbidden. In mercury-cell CAPs, liquid Hg forms a flowing layer at the cathode which conducts an electric current during the electrolysis of NaCl-saturated brines, producing chlorine and sodium hydroxide (caustic soda; Castner–Kellner process). Different Hg emission sources are related to CAPs impacting water, soil, and air [16,17,18]. Even if industrial processes are moving toward Hg-free membrane technologies, Hg cell processing is still extensively used, and the legacy of chlor-alkali activities on ecosystems may persist for different durations, even after closure of the plants [19,20]. In particular, mercury that accumulates in soil undergoes a wide array of transformation processes and redox reactions [21,22]. Mercury released as divalent Hg(II) in the water–soil system may be sorbed by organic matter [23,24]; it can also form Mn-Fe oxy-hydroxides [25] or be absorbed into clays, or it can be reduced to Hg(0) via biotic processes and/or abiotic pathways, such as by Fe(II)-bearing minerals and sulfide phases [26]. Due to its volatilization properties, the Hg(0) that forms may diffuse within the soil and cross the soil surface into the atmosphere [27]. Hg vapor may then be sorbed by soil and transformed into Hg(II). The conversion of inorganic Hg(II) to MeHg in the soil is facilitated by microorganisms, primarily under reducing conditions [28,29,30,31,32,33], and, subsequently, demethylation may occur through abiotic and biotic processes [34]. The complex biogeochemical cycling of mercury is critical in evaluating potential health risks [35]. Indeed, mercury toxicity and bioavailability in the soil depend not only on the total element concentration, but also on its chemical speciation [36]. Speciation analysis is commonly based on selective sequential chemical extraction methods [37], e.g., extraction processes using a specific solvent, X-ray absorption analysis [38], or thermal desorption [39,40,41]. In the present study, the Hg legacy related to a decommissioned solution mine and chlor-alkali plant in southern Tuscany (Italy) was investigated, and a health risk assessment was performed on the basis of a Hg speciation analysis of soil.

## 2. Materials and Methods

### 2.1. Site Description

The study area is located in the Volterra Basin, near the Saline di Volterra village, about 5 km southwest from the Volterra Municipality (southern Tuscany, Italy) (Figure 1). The basin comprises a sedimentary succession [42] that includes upper Messinian evaporites, consisting of gypsum, conglomerates, sandstone, claystone, and halite. Evaporites form a salt deposit (Saline di Volterra Formation) with thickness ranging from a few meters to nearly 200 m., where halite represents about 40% of the total volume [43]. The whole sequence is unconformably overlaid by Pliocene marine clays. The Saline di Volterra salt rock field has been extensively exploited by long-lasting salt solution mining activities, exerting a number of environmental pressures [44], including subsidence, collapses with the creation of sinkholes [45], and the depletion of water resources. In solution mining, a dissolving fluid (generally water) is pumped through boreholes at depth to dissolve the salt. In the study area, NaCl-saturated brine was channeled through pipelines to the Hg-cell chlor-alkali plant at Saline di Volterra. At the end of the industrial process, the exhausted, Hg-bearing NaCl solution was treated with a cation exchange resin to remove part of the mercury and then pumped back into the rock salt field at a depth between 150 and 300 m. About 30 m^3^/h of brine was processed [46]. At the time, the Hg-contaminated wash waters from the cells were discharged into a tributary of the Cecina River (Figure 1) [46], causing high levels of contamination in the aquatic ecosystem. In 1994, a leak in the reinjection pipeline in one of the mining concessions caused the outflow of Hg-rich brine and contamination of the soil in the Piano della Canova locality, an area characterized by ponds formed by collapses derived from the solution mining activity (Figure 1). It was then recognized that the mining activity had had a high environmental impact, and the plant was permanently decommissioned in 1994. A soil phytoremediation project in this area was started in 2016 [47].

### 2.2. Soil Sampling and Analysis

During the year 2000, a survey was conducted to assess the legacy-Hg contamination at the Piano della Canova site, sampling the topsoil (0–30 cm depth) at 22 stations (see Figure 2). A more detailed survey was carried out in 2009, sampling at the same 22 stations in order to monitor changes in Hg concentrations, and at 23 additional sampling points at two depths in the soil profile (0–30 cm and 30–50 cm) (SCL Italy, Milan, Italy, Larderello Group, unpublished data). The present-day soil contamination was investigated through a sampling campaign carried out in 2023, when 26 topsoil samples (0–30 cm depth) in five areas characterized by different degrees of contamination were collected. Sampling was performed by drilling holes using a bucket auger, 10 cm in diameter; samples were stored in polyethylene bags and kept in the dark until laboratory analysis. Before analysis, soils were air-dried to reduce Hg(0) losses [48] and sieved to 2 mm, as required by the guidelines for soil analysis. The total mercury in the soil samples collected during the 2000 and 2009 surveys (SCL Italy, Larderello Group, unpublished data) and in 2023 was determined using a Milestone DMA-80 (US EPA Method 7473). The Relative Standard Deviation (RDS) and accuracy were 5%, on the basis of repeated analyses based on NIST 2711a and ERM-CC018 standards. The results are expressed in mg/kg dry weight. Mercury speciation was quantified considering methyl-Hg (MeHg), water soluble Hg (as chlorides), and volatile Hg (as Hg(0)), which are of relevance for evaluating the pathways associated with potential health risks. In particular, the MeHg analysis was performed using the extraction scheme proposed by Bosze et al. [49] and approved by the Agency for Environmental Protection of the Tuscany region. The water-soluble fraction was estimated using a soil:water ratio of 1:100 for 24 h [50], employing Milli-Q water. The Hg(0) fraction in the soil was estimated by thermal desorption following the method proposed by Ghezzi et al. [41]. Soil pH was determined according to the US EPA Method 9045D, and electrical conductivity (EC) was measured on saturation extracts calculated at 25 °C (HRN ISO 11265) [51].

### 2.3. Environmental Indices, Exposure Models, and Risk Assessment

The extent of soil mercury pollution was evaluated for samples collected during the 2023 survey by using the Contamination Factor (CF) and Geoaccumulation Index (I_geo_). These indices were calculated on the basis of a comparison between the concentration measured in the sample and the background value of 0.2 mg/kg, obtained for samples collected outside the mining area. This value is in the range reported by Dall’Aglio [52] for the background value of southern Tuscany (0.2–0.3 mg/kg in stream sediments).

The CF is a ratio between the element concentration at the sampling site and the background/baseline value or reference value [53]. The CF was calculated as follows:$$CF=\frac{C_{Hg}}{B_{Hg}}$$
where C_Hg_ is the Hg concentration at the sampling site and B_Hg_ is the background or reference concentration of Hg at the site. According to the CF value, four contamination levels can be identified: CF < 1 (low contamination), 1 < CF < 3 (moderate contamination), 3 < CF < 6 (considerable contamination), and CF > 6 (very high contamination).

I_geo_ was calculated using the following formula [54]:$$I_{geo}={Log}_{2}\frac{C_{Hg}}{1.5*B_{Hg}}$$
where C_Hg_ is the measured concentration for Hg in the examined sample and B_Hg_ is the background value measured for the soil outside the mining area. I_geo_ can be categorized into seven classes: Class 0 (I_geo_ < 0, unpolluted); Class 1 (0 < I_geo_ < 1, lightly polluted); Class 2 (1 < I_geo_ < 2, moderately polluted); Class 3 (2 < I_geo_ < 3, moderately severely polluted); Class 4 (3 < I_geo_ < 4, severely polluted); Class 5 (4 < I_geo_ < 5, severely extremely polluted); and Class 6 (I_geo_ > 5, extremely polluted).

A risk analysis was performed following the Risk Based Corrective Action procedure, which applies the deterministic approach outlined in the ASTM standards [55] and in the United States Environmental Protection Agency (US EPA) guidelines [56,57,58]. The selected exposure routes were surface soil ingestion, dermal contact, soil dust, and outdoor vapor inhalation in a residential setting. Human receptors were both adults and children. The risk for both non-carcinogenic (Hazard Quozient-HQ) and chronic effects in humans was calculated [59] for each exposure pathway using the Risk.net software (version 3.1.1 pro; http://www.reconnet.net/Software.htm, accessed on 30 April 2024). The chronic daily intake (CDI, mg/kg/day), representing the exposure to a toxic agent averaged over a long period of time through ingestion (CDI_ing_) or dermal contact (CDI_derm_), was derived from standardized sets of equations [57,60], as follows:$${CDI}_{ing}=C_{POE}*\frac{{IR}_{ing}*EF*ED}{BW*AT}*10^{-6}$$
$${CDI}_{derm}=C_{POE}*\frac{SA*SAF*ABS*EF*ED}{BW*AT}*10^{-6}$$
where C_POE_ is the exposure point concentration of the contaminant in soil (mg/kg), equal to the concentration at the source (C_soil_) in the case of direct exposure pathways; 10^−6^ is a conversion factor (kg/mg). For the remaining parameters, recommended values were used (Table 1).

In the case of direct ingestion and dermal contact, the non-carcinogenic Hazard Quotient HQ (i.e., HQ_ingestion_ and HQ_dermal_) was calculated by dividing the chronic daily intake by the corresponding Reference Dose (RfD, mg/kg/day), defined as the maximum daily exposure to a toxic agent that would not produce any appreciable deleterious effect on human health (for reference toxicity values, see Table 2):$$HQ=\frac{CDI}{RfD}$$

For the dust inhalation pathway [61], HQ_inhalation_ was calculated by dividing the exposure concentration (EC, mg/m^3^) by the reference toxicity concentration value (RfC, mg/m^3^), which represents an estimate of continuous inhalation exposure without appreciable risk of deleterious effects during a lifetime:$$HQ=\frac{EC}{RfC}$$

RfC for elemental mercury was set to 3.0 × 10^−4^ mg/m^3^ [58] (Table 2) and EC was estimated starting from the predicted concentration in air at the exposure point (C_POE_, mg/m^3^), according to:$$EC=\frac{C_{POE}*ET*EF*ED}{AT}$$

For the ET, EF, ED, and AT parameters, recommended values were used (Table 1). The concentration in air at the exposure point (C_POE_, mg/m^3^) was calculated starting from the concentration at the source of emission (C_soil_) and applying specific “fate&transport” (FT) analytical models (C_POE_ = C_soil_ × FT) related to the dust and vapor inhalation pathway, respectively, as defined by the ASTM standard [55].

In particular, the transport model related to the inhalation of the contaminant adsorbed on breathable particles formed by wind erosion may be identified using the particle emission factor (VFp, derived from [62]), while in the case of the vaporization of a compound from the surface soil to outdoor air, FT is the volatilization factor (VF_SS_, derived from [63]). Both VFp and VF_SS_, which predict the attenuation of a chemical of concern away from the soil source, were calculated (see Table 1 for values) using both site-specific data (e.g., 80 m as the width of the source area parallel to the wind direction, silt loam (according to U.S. Department of Agriculture soil texture classification), and 2.05 m/s as the ambient air velocity in the mixing zone) obtained during field investigation and the conservative default values suggested by the ASTM standard.

The total non-carcinogenic risk for a single substance defines the screening level individual Hazard Index (HI):$$HI={HQ}_{ingestion}+{HQ}_{dermal}+{HQ}_{inhalation}$$

A HI value lower than unity indicates an acceptable risk [59].

The maximum allowed concentration of contaminants in soil, intended to be protective of human health (site-specific soil screening levels-SSLs, according to US EPA guidelines), was obtained by following the Risk Based Corrective Action procedure [55,56]. In this approach, the exposure equations and pathway models used for estimating the potential adverse effects are run in reverse to back-calculate the acceptable level of a contaminant concentration in soil corresponding to a target level of risk [56,57]. Risk-based SSLs for the different outdoor exposure pathways and for a residential setting (Table 2) were derived from standardized sets of equations based on the updated U.S. Environmental Protection Agency’s human health risk assessment methods [64].

**Table 1 toxics-12-00436-t001:** Recommended exposure factors [65,66] and FT values [55].

Symbol	Definition (units)	Value
**IR_ing_**	Ingestion rate (mg/day) (accounting for both soil and dust ingestion)	100 for adult, 200 for children
**EF**	Exposure frequency (day/year) for a residential setting	350
**ET**	Exposure time (h/day) for a residential setting	24
**ED**	Exposure duration (years) for a residential setting	24 for adults, 6 for children
**SA**	Exposed skin area (cm^2^)	5700 for adults, 2800 for children
**SAF**	Skin adherence factor (mg/cm^2^)	0.07 for adults, 0.2 for children
**ABS**	Dermal absorption factor (fraction of contaminant absorbed dermally from soil, unitless)	0.01 (chemical specific)
**BW**	Average body weight (kg)	70 for adults, 15 for children
**AT** **(Ingestion and dermal contact)**	Average time of exposure (day)	ED × 365 day/year for non-carcinogens
**AT (inhalation)**	Average time of exposure (h)	ED × 365 day/year × 24 h/day for non-carcinogens
**VFp**	Total respirable particulate concentration from the soil source (mg/m^3^-air/mg/kg-soil)	1.35 × 10^−11^
**VFss**	Volatilization factor, subsurface soil to ambient air (mg/m^3^-air/mg/kg-soil)	1.32 × 10^−5^

**Table 2 toxics-12-00436-t002:** Summary of Hazard Quotient (HQ) for non-carcinogenic risk calculated for different exposure routes for children and adult receptors (values for adults, when different, are in brackets) based on total mercury maximum concentration and speciation data (see text). RfD_derm_ was assumed to be equal to RfD_oral_ because it is not necessary to apply any “gastrointestinal absorption factor” to adjust the available oral toxicity values [57]. * RfC not available; ** US UT [60] and OEHHA [67]; *** US EPA [58].

Hg_tot_ = 25.5 mg/kg Hg(0)= 4.4 mg/kg MeHg = 0.3 μg/kg Mercuric Chloride = 1.3 μg/kg	HQ	Soil Ingestion	Dermal Contact	Inhalation Vapor	Inhalation Dust	Sum of Outdoor Exposures (HI)	SSL
**Elemental mercury** RfD_ing_ = 1.6 × 10^−4^ mg/kg/day (**) RfD_derm_ = RfD_oral_ RfC = 3 × 10^−4^ mg/m^3^ (***)		0.352 (3.77 × 10^−2^)	9.84 × 10^−3^ (1.50 × 10^−3^)	0.186	1.89 × 10^−7^	0.547 (0.225)	8.04 (19.6)
**Methylmercury** RfD_ing_ = 10^−4^ mg/kg/day (***) RfD_derm_ = RfD_oral_		3.84 × 10^−5^ (4.11 × 10^−6^)	1.07 × 10^−6^ (1.64 × 10^−7^)	-	*	3.94 × 10^−5^ (4.27 × 10^−6^)	7.61 (70.2)
**Mercuric Chloride** RfD_ing_ = 3 × 10^−4^ mg/kg/day (***) RfD_derm_ = RfD_oral_		5.54 × 10^−5^ (5.94 × 10^−6^)	1.55 × 10^−6^ (2.37 × 10^−7^)	-	*	5.70 × 10^−5^ (6.17 × 10^−6^)	22.8 (211)
**Hg_tot_ attributed to Hg(0)**		2.04 (0.218)	5.71 × 10^−2^ (8.71 × 10^−3^)	1.09	1.11 × 10^−6^	3.18 (1.31)	
**Hg_tot_ attributed to MeHg**		3.26 (0.349)	9.13 × 10^−2^ (1.39 × 10^−2^)	-	*	3.35 (0.363)	
**Hg_tot_ attributed to Mercuric Chloride**		1.09 (0.116)	3.04 × 10^−2^ (4.65 × 10^−3^)	-	*	1.12 (0.121)	

## 3. Results

### 3.1. Geochemical Parameters and Total Mercury

The soil pH was alkaline, varying between 7.9 and 9.8, and generally increasing with depth; EC was quite heterogeneous, varying in the range 0.2–1.5 dS/m. The same variability characterized the different surveys, and no correlation was observed among soil parameters. The Hg concentrations measured in the topsoil (0–30 cm depth) during the 2000, 2009, and 2023 surveys are graphically shown in Figure 2. The main descriptive statistic parameters of the three sampling campaigns (0–30 cm depth) are reported in Table 3 (see Appendix A for the complete database). The concentrations at 30–50 cm depth, recorded during the 2009 survey, are reported in Appendix A (Appendix A). The data show that, in 2000, mercury contamination in the topsoil ranged between 0.1 to 334 mg/kg; by 2009, the Hg concentration had decreased, ranging between 0.1 and 41 mg/kg, except at one station, where a concentration as high as 258 mg/kg was measured. Furthermore, the data show that in 2009, the contamination was lower at higher depth, ranging between 0.1 and 21 mg/kg at 30–50 cm, being below 1 mg/kg in most samples. The Hg content found in soil in the 2023 survey ranged between 0.1 and 25.5 mg/kg; for seven samples out of 26, Hg exceeded the 1 mg/kg concentration threshold imposed by Italian regulations for soil mercury in residential and recreational areas. Although a drastic reduction has occurred over the last decade, the data highlight the legacy of the contamination almost thirty years after the salt-sludge emission.

### 3.2. Mercury Speciation

A speciation analysis for risk assessment applied to the most contaminated soil sample collected during the 2023 survey (total Hg: 25.5 mg/kg) indicated that water soluble Hg and MeHg represented minor fractions (1.3 µg/kg and 0.3 µg/kg, respectively) of the total mercury content. The thermal desorption profile, obtained by heating the most contaminated soil sample at 100 °C and measuring the desorbed Hg vapor over time, is shown in Figure 3. The total emission can be entirely attributed to the release of Hg(0). The kinetics of Hg desorption from soil was modeled using the following exponential equation, as suggested for the kinetics of Hg(0) desorption from soil [68]:(1)$$Hg(t)=a\left(1-e^{-kt}\right)$$
where *a* and *k* are the asymptotic value (corresponding to the initial Hg(0) concentration in soil) and the kinetic rate constant, respectively.

The fitted parameters, obtained with the Origin 8.0 software (OriginLab Corp, Northampton, MA, USA), are *a* = 4420 ± 57 ng/g and *k* = 0.0134 ± 0.0003 h^−1^, with a correlation coefficient r^2^ = 0.99.

### 3.3. Mercury Risk Assessment

According to the CF calculation (Figure 4), seven samples collected during 2023 had low contamination, five were moderately contaminated, and seven were severely contaminated.

According to the I_geo_ classification, half of the collected samples could be considered uncontaminated (I_geo_ < 0), while two ranged from uncontaminated to moderately contaminated (I_geo_ 0–2) soils. Five samples were from moderately to strongly contaminated and only two samples could be considered extremely contaminated.

A health risk assessment, based on typical human exposure assumptions and standard toxicological guidance values, was performed considering both the highest value of mercury content and the obtained speciation results and taking into account the US EPA guidelines (elemental mercury (Hg(0)), methylmercury (MeHg) and soluble mercuric chloride, respectively), as shown in Table 2.

US EPA guidelines do not provide any oral toxicity reference value for elemental mercury. Therefore, soil ingestion/dermal contact pathways were evaluated using the oral reference toxicity value proposed by OEHHA [68] and reported by US UT [69].

Hazard Quotients related to the highest mercury concentration associated with soil ingestion were higher than the acceptance threshold for children in a residential setting. In addition, the vapor inhalation pathway has given rise to potential adverse health effects. The risk was generally acceptable in all the other cases.

The soil screening levels (SSLs) related to Hg(0), methylmercury, and mercuric chloride, respectively, were also assessed. It can be observed that the obtained screening values, which were higher than the thresholds imposed by Italian regulations for residential soil (1 mg/kg), were significantly lower than the highest Hg concentration measured in the soil but higher compared with the actual soil concentrations of the corresponding Hg-fraction.

## 4. Discussion

The mercury used in the electrolytic cell of the chlor-alkali process and discharged into the floodplain soil through a brine-transporting pipeline leak in 1994 was mainly metallic mercury Hg(0). Biester et al. [16] reported that the Hg(0) initially deposited in soils by dispersion from a chlor-alkali plant was quickly volatilized and slowly oxidized to Hg(II) through both chemical and biological processes. Indeed, Hg(0) released into soil can be readily converted to vapor at ambient temperature due to its low latent heat of evaporation, and the vapor can be sorbed by soil [70]. Mercuric forms may strongly interact with soil constituents in different environmental conditions, both by sorption and by the formation of stable complexes with different ligands, also considering the Hg(II)-natural organic matter affinity (e.g., [22]). In fact, mercuric ions may become relatively immobile in clayed soil layers [71] and in humic acid-rich topsoil. In addition, Hg(II) may react with sulfur, leading to the formation of the sparingly soluble cinnabar and metacinnabar. This process inhibits Hg(II) methylation and further immobilizes mercury in soils.

A comparison of the total Hg concentration patterns in soil in 2000, 2009, and 2023 shows that pollution decreased over time. However, the EF and I_geo_ values indicate that in 2023, Piano della Canova soils were still enriched in Hg by factors of up to 6.41 and 127.5, respectively, compared to local background levels, reflecting the long retention time for this contaminant. Leaching tests indicated that the legacy Hg in the soil has a low tendency to be mobilized by water, and hence, subsurface migration by infiltrating rain or during flooding was reduced. Indeed, the Hg content measured both in groundwater and ponded sinkholes ranged between <0.05 and 0.07 µg/L (data from: SCL Italy, Larderello Group, unpublished data). Speciation analyses indicated that a minor fraction of Hg(0) from the pristine contamination was present in soil in 2023, representing only 17.3% (4.4 mg/kg) of the total Hg measured. The obtained kinetic constant was lower than the kinetic constant reported by Park et al. [67] for Hg(0) degassing from soil. This suggests that Hg(0) was well retained in the soil as matrix bound Hg(0).

Even if the Hg(0) content in soil is difficult to determine unequivocally, the data indicated gaseous mercury fluxes after contamination [72]. Mercury volatilization from soil and emission rates might have increased during flooding events that have characterized the area, increasing the soil pore water content and the Hg(0) concentration in soil air [73]. Furthermore, it must be considered that flooding induces reducing conditions in soil, with the possible formation of mercury sulfide (HgS) [74]. Detailed investigations on Hg partitioning and speciation changes in soil are beyond the scope of this study, which focuses on the different Hg species only for a risk assessment. However, it is probable that the Hg still present in the Piano della Canova soil is mostly bound to organic matter and/or contained in sulfide minerals [75].

Our risk assessment indicated that the major risk to health associated with mercury is through soil ingestion (Figure 5). Generally, vapor inhalation does not produce significant health risks, and dermal contact and dust inhalation routes give negligible HQ values.

In summary, the highest measured total mercury concentration in the surface soil (25.5 mg/kg), if entirely attributed to Hg(0), MeHg, or mercuric chloride, always yielded a total outdoor hazard index that exceeded unity for non-cancer risk on humans (especially children).

In contrast, our risk assessment based on the actual speciation data indicated an acceptable risk for all the exposure pathways considered, including soil ingestion (Figure 5).

It is worth mentioning that the speciation data obtained in this study allowed us to undertake a realistic risk assessment, avoiding overestimations.

All SSLs were found to be lower than the total mercury concentration measured. It is worthy of note that, if speciation data were not available, this would lead us to consider remedial actions. Instead, our comparison of the risk threshold values with the corresponding Hg-fractions led to a different outcome, and remedial actions were found to be unnecessary.

In terms of repurposing the land for recreational use, the soil ingestion pathway related to total mercury would produce a human health risk only for exposure frequencies over 105 days/year.

Even if no indoor environment is currently planned to be realized on site, the migration of Hg(0) vapor through basements should also be considered, since it too might contribute to the human health risk. If both the default building parameters and speciation data are considered, and the appropriate transport factor model suggested by ASTM standards [76] is used, a significant risk is calculated.

## 5. Conclusions

In this study, the spatial and temporal mercury contamination in soils in a decommissioned solution mining area and chlor-alkali plant in southern Tuscany was addressed, and a health risk assessment was performed.

Despite the fact that the total Hg concentration generally decreased from 2000 to 2023 and was mostly confined to the topsoil layer, almost thirty years after the contamination event, legacy-Hg exceeding the 1 mg/kg threshold and reaching 25.5 mg/kg was detected in some soil samples. According to our speciation analysis, Hg(0) represented about 17.3% of the total Hg in the most contaminated samples, and the water soluble and organic Hg fractions were not significant. These results indicate that present-day Hg(0) represents a minor fraction of the original metallic Hg source, and that the extent of Hg transport in the subsurface through infiltrating water is negligible, preventing offsite movement of the contaminant. These observations have important implications for the long-term management of the area.

The health risk was assessed using both the highest value of total mercury content and the actual speciation results. In the first case, the major threat for health associated with mercury is through the soil ingestion pathway. On the other hand, a risk assessment based on the actual speciation data indicated an acceptable risk for all the outdoor exposure pathways considered, including soil ingestion. This represents a basis for repurposing the formerly contaminated site as a public leisure area.

Furthermore, this study demonstrates the usefulness of site-specific investigations, using geochemical data for modelling the Hg-risk associated with human exposure. Site-specific risk assessments hence represent a useful decision-making tool concerning the cleanup of contaminated sites, avoiding waste of public money.

## Figures and Tables

**Figure 1 toxics-12-00436-f001:**
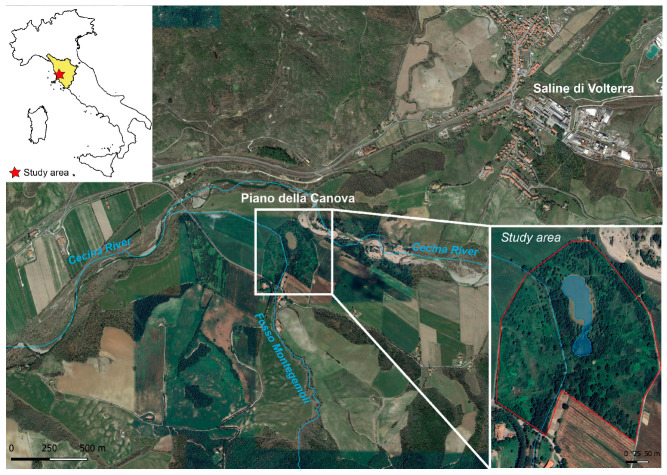
Location of the study area (basic map: Google Maps^®^ 2024).

**Figure 2 toxics-12-00436-f002:**
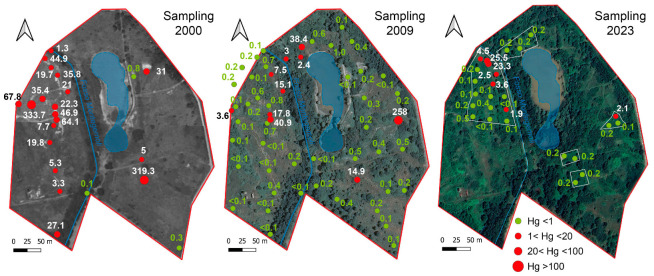
Location of the sampling point and Hg concentrations in topsoil (0–30 cm depth) measured during the 2000, 2009, and 2023 surveys.

**Figure 3 toxics-12-00436-f003:**
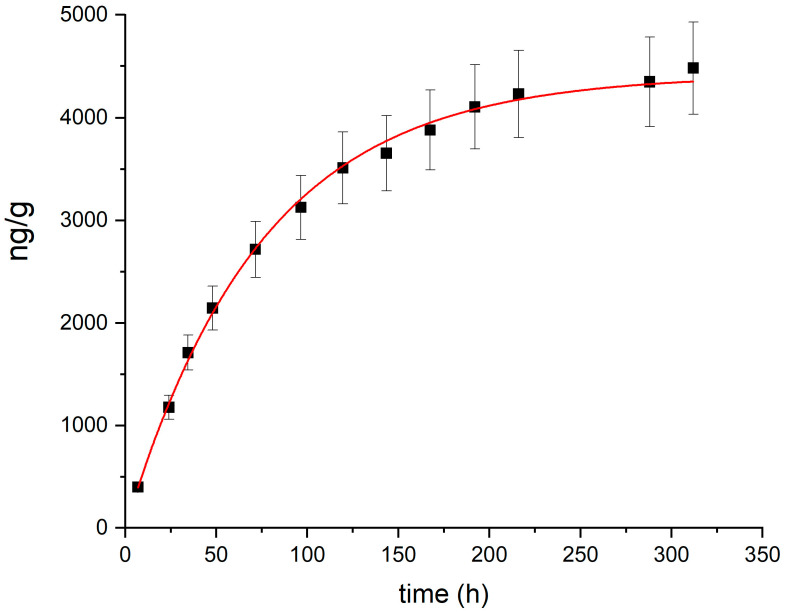
Thermal desorption profile (black squares) and fitting (red line) for the most contaminated soil sample collected during the 2023 survey.

**Figure 4 toxics-12-00436-f004:**
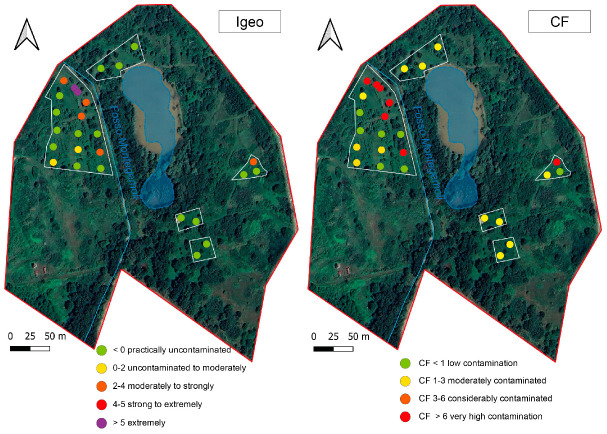
I_geo_ and CF calculated for samples collected during the 2023 survey.

**Figure 5 toxics-12-00436-f005:**
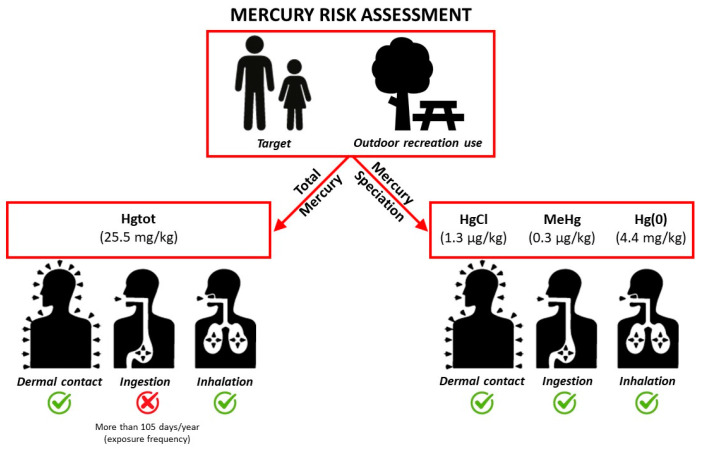
Health Risk Assessment.

**Table 3 toxics-12-00436-t003:** Descriptive parameters of the three sampling campaigns (0–30 cm depth).

Hgtot (mg/kg)	2000 Sampling	2009 Sampling	2023 Sampling
n°	22	53	24
Min	0.13	0.10	0.10
Max	334	258	25.5
Mean	50.5	7.8	2.8
Median	21.7	0.2	0.2
SD	91.6	36	6.8

## Data Availability

Data are contained within the article and Appendix A.

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
