# Peer review of "The Legacy of Hg Contamination in a Past Mining Area (Tuscany, Italy): Hg Speciation and Health Risk Assessment"

_toxics, 2024, doi:10.3390/toxics12060436_

Round 1

Reviewer 1 Report

Comments and Suggestions for Authors

In this manuscript, the authors provide a thorough investigation of mercury contamination in soils at a decommissioned chlor-alkali plant site in Tuscany, Italy. The study is comprehensive, covering historical context, soil sampling over multiple years, mercury speciation analysis, and health risk assessment. This paper presents novel insights and is worthy of publication. Comments for further revisions are provided below. 

Abstract: My general impression is that there is a clear boundary between the research background and experimental work, which makes the study's objective appear vague. 

Introduction: The introduction should emphasize the ongoing relevance of mercury contamination issues on a global scale. 

Materials and Methods: The historical context and timeline of contamination levels are well-documented. A figure could be added to vividly illustrate the pollution's progression and persistence, along with the temporal sampling campaign. Regarding methodological details, only the measurements of Hg parameters are mentioned. What about the other parameters, such as soil properties? In lines 128-129, the sentence "CHg is the Hg concentration at the sampling site, CHg is the background or reference concentration of the Hg at the site" contains a repetition error and should be corrected. 

Results: Additional interpretation of the data is needed, particularly in relation to environmental factors influencing mercury speciation. 

Discussion: This section describes many environmental processes related to Hg mobility and risks. The paper would be more engaging if there were a figure showing the interconnected processes of Hg biogeochemistry and potential risks. 

Conclusion: The authors should address the potential for future research directions and the practical applications of the study’s findings in environmental management.

Comments on the Quality of English Language

It's generally okay, but can be further improved

Author Response

Dear Reviewer 1,

We thank you for the useful suggestions that greatly helped us to improve our paper. In the revised manuscript, we carefully considered all the comments and suggestions. In the following, we provide a point-by-point explanation of changes made in reaction to each comment. We hope that the revised manuscript is now adequate for publication.

Abstract: My general impression is that there is a clear boundary between the research background and experimental work, which makes the study's objective appear vague.

The abstract has been revised, making clearer the objective of the study, and better stressing the implications of the obtained results.

Introduction: The introduction should emphasize the ongoing relevance of mercury contamination issues on a global scale.

In the Introduction, the global-scale relevance of Hg contamination has been briefly addressed, and a reference has been added (Obrist, D., Kirk, J.L., Zhang, L. et al. 2018. A review of global environmental mercury processes in response to human and natural perturbations: Changes of emissions, climate, and land use. Ambio 47, 116–140. https://doi.org/10.1007/s13280-017-1004-9).

Materials and Methods: The historical context and timeline of contamination levels are well-documented. A figure could be added to vividly illustrate the pollution's progression and persistence, along with the temporal sampling campaign.

We redraw Figure 1 in order to highlight the spatial and temporal variation of Hg concentration.

Regarding methodological details, only the measurements of Hg parameters are mentioned. What about the other parameters, such as soil properties?

The analytical procedure for soil pH has been added in the “2.2 Soil sampling and analysis” section, and the results have been reported in the new section “3.1 Geochemical parameters and total mercury”.

In lines 128-129, the sentence "CHg is the Hg concentration at the sampling site, CHg is the background or reference concentration of the Hg at the site" contains a repetition error and should be corrected.

We correct the typing error. Now CHg is the Hg concentration at the sampling site and BHg is the background or reference concentration of the Hg at the site.

Results: Additional interpretation of the data is needed, particularly in relation to environmental factors influencing mercury speciation.

The Referee is right; unfortunately, detailed environmental information is missing on the site, making difficult to evaluate the possible time-dependent effects on the measured Hg speciation.

Discussion: This section describes many environmental processes related to Hg mobility and risks. The paper would be more engaging if there were a figure showing the interconnected processes of Hg biogeochemistry and potential risks.

We added a new Figure showing the health risk assessment (see Figure 5 into the text).

Conclusion: The authors should address the potential for future research directions and the practical applications of the study’s findings in environmental management.

Further implications of the study have been included in the Conclusions, and the text upgraded accordingly.

Reviewer 2 Report

Comments and Suggestions for Authors

The manuscript reports the legacy of Hg contamination in an abandoned mining area and estimates potential human health risks using the USEPA model. The manuscript is well presented and writing flow was logical. However, I have few minor suggestions, which I believe will increase the readability of the manuscript.

1. Line 107: It would be better if you spell out "RSD", though it is a very common term in quality control and quality assurance.

2. Also, clarify if you are reporting dry weight basis concentration or not, like mg/kg dry weight.

3. Results section is a bit hard to follow up. There are Hg contamination, Hg pollution level in soil, and potential human health risk of Hg. It would be better if the authors subdivide the results under appropriate subheadings.

4. Line 319: HgS could be placed in first bracket.

Author Response

Dear Reviewer 2,

We thank you for the useful suggestions that greatly helped us to improve our paper. In the revised manuscript, we carefully considered all the comments and suggestions. In the following, we provide a point-by-point explanation of changes made in reaction to each comment. We hope that the revised manuscript is now adequate for publication.

  1. Line 107: It would be better if you spell out "RSD", though it is a very common term in quality control and quality assurance.

We spelt out “RDS”, that means “Relative Standard Deviation”.

  1. Also, clarify if you are reporting dry weight basis concentration or not, like mg/kg dry weight.

The concentration was calculated on dry weight. We added this information into the text.

  1. Results section is a bit hard to follow up. There are Hg contamination, Hg pollution level in soil, and potential human health risk of Hg. It would be better if the authors subdivide the results under appropriate subheadings.

We added subheadings into the results section.

  1. Line 319: HgS could be placed in first bracket.

We added the brackets.

Reviewer 3 Report

Comments and Suggestions for Authors

The authors of the reviewed paper presented results regarding some ecological issues related to the leakage dispersion of highly contaminated Hg(0)-bearing brines into the environment. The problem described is very important and probably common to many areas all over the world. The whole study in my opinion is very well designed and the gathered results assessed correctly.

My main concern is the quantitative data, which is not given at all. I would like to see the variability of the results within time & area, apart from their graphical interpretation. How many samples were analyzed in total? What was the SD and variation? How many samples were measured within one sampling point? Where there any other metals studied along with mercury? What are their levels in the top-soil samples? Are there any differences among various soil layers? There was a remark about other soil layers investigated in 2009 (30-50 cm). The authors in this paper should focus not only on the risk factor but also highlight the quantitative data. Please provide some extra information (plus the table with the results) connected with the issues mentioned to complete this interesting paper. 

Author Response

We thank you for the useful suggestions that greatly helped us to improve our paper. In the revised manuscript, we carefully considered all the comments and suggestions. In the following, we provide a point-by-point explanation of changes made in reaction to each comment. We hope that the revised manuscript is now adequate for publication.

My main concern is the quantitative data, which is not given at all. I would like to see the variability of the results within time & area, apart from their graphical interpretation. How many samples were analyzed in total? What was the SD and variation? How many samples were measured within one sampling point?

We added a new table into the text (see Table 2 into the text) with the statistical parameters (min, max, SD, median, average). You can find the complete database into the supplementary materials (new Table S1).

Where there any other metals studied along with mercury? What are their levels in the top-soil samples? Are there any differences among various soil layers?

Unfortunately, only the mercury concentration was determined. At the moment, we do not have a sufficient sample quantity to perform other analyzes and integrate the data.

There was a remark about other soil layers investigated in 2009 (30-50 cm).

We described the subsoil (30-50 cm) at the lines 229-230 and the data are graphically reported into Figure S1 (supplementary materials).

The authors in this paper should focus not only on the risk factor but also highlight the quantitative data. Please provide some extra information (plus the table with the results) connected with the issues mentioned to complete this interesting paper.

We added a new Table 2 and Table S1 (supplementary material) in order to highlight the quantitative date. We also slightly modified the text.